# Chronic Inflammation Disrupts Circadian Rhythms in Splenic CD4+ and CD8+ T Cells in Mice

**DOI:** 10.3390/cells13020151

**Published:** 2024-01-13

**Authors:** Misa Hirose, Alexei Leliavski, Leonardo Vinícius Monteiro de Assis, Olga Matveeva, Ludmila Skrum, Werner Solbach, Henrik Oster, Isabel Heyde

**Affiliations:** 1Institute of Neurobiology, University of Lübeck, 23562 Lübeck, Germany; misa.hirose@uni-luebeck.de (M.H.); l.skrum@uni-luebeck.de (L.S.); 2Center of Brain, Behavior and Metabolism, University of Lübeck, 23562 Lübeck, Germany; 3Institute of Experimental Dermatology, University of Lübeck, 23562 Lübeck, Germany; 4T-Knife GmbH, 13125 Berlin, Germany; 5Institute for Medical Microbiology and Hygiene, University of Lübeck, 23562 Lübeck, Germany; werner.solbach@uni-luebeck.de

**Keywords:** CD4^+^ T cells, CD8^+^ T cells, circadian clock, experimental autoimmune encephalomyelitis (EAE), inflammation, pertussis toxin (PTx), complete Freund’s adjuvant (CFA), spleen

## Abstract

Internal circadian clocks coordinate 24 h rhythms in behavior and physiology. Many immune functions show daily oscillations, and cellular circadian clocks can impact immune functions and disease outcome. Inflammation may disrupt circadian clocks in peripheral tissues and innate immune cells. However, it remains elusive if chronic inflammation impacts adaptive immune cell clock, e.g., in CD4+ and CD8+ T lymphocytes. We studied this in the experimental autoimmune encephalomyelitis (EAE), a mouse model for multiple sclerosis, as an established experimental paradigm for chronic inflammation. We analyzed splenic T cell circadian clock and immune gene expression rhythms in mice with late-stage EAE, CFA/PTx-treated, and untreated mice. In both treatment groups, clock gene expression rhythms were altered with differential effects for baseline expression and peak phase compared with control mice. Most immune cell marker genes tested in this study did not show circadian oscillations in either of the three groups, but time-of-day- independent alterations were observed in EAE and CFA/PTx compared to control mice. Notably, T cell effects were likely independent of central clock function as circadian behavioral rhythms in EAE mice remained intact. Together, chronic inflammation induced by CFA/PTx treatment and EAE immunization has lasting effects on circadian rhythms in peripheral immune cells.

## 1. Introduction

Endogenous circadian clocks regulate daily rhythms of behavioral and physiological processes in vertebrate and invertebrate species [1]. At the molecular level, these clocks are based on interlocked transcriptional–translational feedback loops (TTFLs) that ensure robust control over rhythmic gene expression programs [2]. Transcriptional activators such as brain and muscle ARNT-like 1 (BMAL1 or ARNTL) together with circadian locomotor output cycles kaput (CLOCK) regulate the rhythmic expression of hundreds of target genes via binding to enhancer boxes (E-boxes). CLOCK:BMAL1 heterodimers induce the expression of *period 1-3* (*Per1-3*), *cryptochrome 1/2* (*Cry1/2*), *nuclear receptor subfamily 1*, *group D*, *member 1/2* (*Nr1d1/2*, encoding reverse-erythroblastosis virus α/β (REV-ERBα/β)), *retinoic acid receptor-related orphan receptors α/β/γ* (*Rorα/β/γ*), as well as the *transcription factor albumin D site-binding protein* (*Dbp*) (reviewed in [1,2]). PER1-3 and CRY1/2 accumulate over the time, form heterodimers, and translocate into the nucleus, where they interact with CLOCK:BMAL1, repressing their own transcription forming the core TTFL. In an accessory loop, REV-ERBα/β and RORα/β/γ repress or induce the expression of *Bmal1*, respectively. In a second auxiliary loop, DBP activates, whereas E4 promotor-binding protein 4 (E4BP4) represses the transcription of *Per1-3*, *Rev-Erbα/β*, and *Rorα/β/γ*. Moreover, circadian rhythms are further regulated by chromatin remodeling, post-transcriptional, and post-translational modifications (reviewed in [3,4,5]). 

At the organismal level, the circadian timing system is organized in a hierarchical manner, with a light-entrainable central circadian pacemaker residing in the hypothalamic suprachiasmatic nucleus (SCN) [1]. The SCN clock controls the rhythmicity of neural, hormonal, and metabolic signals responsible for the synchronization of cellular clocks throughout the body [6,7,8]. In mammals, differentiation, migration, and function of immune cells display daily oscillations [9,10,11,12]. T lymphocytes, for example, show daily rhythms in migration and function [9,13,14,15] and harbor intrinsic circadian clocks [16,17,18] affecting T cell immunity, e.g., in EAE [19]. Immune cell maturation and function partly depend on cell-intrinsic clocks, but circadian systemic cues and microenvironment, provided by the clock network, are also needed for efficient immune cell differentiation and action [17,18,19,20,21,22,23]. 

In a commonly used mouse model, EAE is induced by immunization with myelin oligodendrocyte glycoprotein (MOG) peptide 35–55 in complete Freund’s adjuvant (CFA) followed by injections of pertussis toxin (PTx) to boost immune reactions and permeabilize the blood–brain barrier so that activated lymphocytes can reach cells in the central nervous system (CNS) [24,25]. CFA induces chronic peripheral inflammation and pro-inflammatory cytokine production in the CNS [26,27,28]. In EAE, T cells accumulate and are activated in the draining lymph nodes and finally migrate to the CNS, where they attack oligodendrocytes, which results in the clinical signs of the disease [29]. While disruption of the circadian clock system is known to result in immunological dysfunction aggravating diseases, e.g., neurodegeneration, muscle atrophy, and arthritis [30,31,32,33], there is growing evidence for a feedback of inflammation on clock function [34,35]. For example, chronic inflammation disrupts circadian rhythms in various immune cells such as macrophages, neutrophils, and mononuclear cells, but also in cells of peripheral organs like liver, colon, and adipose tissue [36,37,38,39,40,41,42,43]. However, it remains elusive if circadian clock disruptions are also apparent in tissue resident cells of the adaptive immune system. 

We hypothesized that—similar to what had been observed for innate immune cells—a chronic inflammatory state, as induced by EAE immunization, disrupts transcriptional circadian rhythms in splenic CD4+ and CD8+ T cells. We found that not only EAE immunization but also CFA/PTx treatment disrupted T cell circadian clocks, while immune cell expression was mostly impacted in a non-circadian manner.

## 2. Materials and Methods

### 2.1. Animals and Immunization

Female C57BL/6J mice were obtained from Janvier Labs (Le Genest-Saint-Isle, France) or bred in the animal facility of the University of Lübeck. Mice had access to food and water ad libitum. All procedures were ethically assessed and approved by the local state authorities (MELUND Schleswig-Holstein, Germany, AZ (52-4/13) and (74-7/19)). Mice were group housed (2–4 animals per cage) under a 12 h light:12 h dark cycle (LD; 200 lux in the light phase). EAE induction was conducted according to the manufacturer’s protocol (#EK-2110, Hooke Laboratories, Lawrence, MA, USA) during the light phase (Zeitgeber time (ZT) 7–9; “lights on” at ZT0) at 9–13 weeks of age. Briefly, mice were anesthetized (100 mg/kg ketamine (WDT, Garbsen, Germany) and 10 mg/kg xylazine (Bayer, Leverkusen, Germany) i.p.) and bilaterally injected s.c. with 100 µg of MOG peptide fragment 35–55 emulsified in 100 µL CFA (10 mg/mL M. tuberculosis in complete Freund’s adjuvant, Hooke Laboratories) per side, followed by i.p. injection of 400 ng pertussis toxin (PTx, lot 1004) in phosphate-buffered saline (PBS, Gibco, Paisly, UK) immediately and 24 h later (EAE group). A second cohort of age-matched mice was anesthetized with ketamine/xylazine (as EAE animals) and received bilaterally s.c. injections of 100 µL CFA (10 mg/mL M. tuberculosis in CFA) without MOG (Hooke Laboratories) followed by PTx (250 ng, lot 1008, #CK-2110, Hooke Laboratories) i.p. injections at comparable times to the EAE mice (CFA/PTx group). PTx concentrations were adjusted on the basis of lot potency comparison studies provided by the manufacturer. A third group of age-matched animals was left undisturbed (untreated, CON group). All mice were checked daily and, from day 7 on, signs of paralysis were scored (around ZT2–4) as described [19]. Animals with hind limb paralysis (scores ≥ 2) were provided with additional food and water at the bottom of their cage. A subset of EAE (*n* = 4 cages) mice was transferred to constant darkness (DD) conditions for circadian activity rhythm analysis starting at day 2 after the immunization and compared to naïve mice. 

### 2.2. Activity Analysis

Activity was recorded using passive infrared detectors placed on top of the home cages starting at least 4 d prior to EAE induction. Different sets of animals were used for activity recordings and tissue collection for subsequent gene expression analysis. Activity data were analyzed using ClockLab Analysis software, version 6.1.02 (Actimetrics, Evanston, IL, USA). The average activity during the four pre-experimental days was used as the baseline for further analysis. Relative light phase activity was calculated daily. To investigate central clock period lengths, χ2 periodogram analysis was performed on the DD activity data.

### 2.3. Tissue Collection and Preparation

Mice were released into DD during the chronic phase of EAE (day 23–25 after immunization, stable EAE score) and sacrificed by cervical dislocation on day 2 at 6 h intervals (37–55 h after “lights off”, *n* = 2–3 per time point). A CON group of non-immunized mice, kept under the same conditions, was sacrificed in parallel (*n* = 3–4 per time point). A group of CFA/PTx-treated mice were sacrificed at comparable days as EAE-induced mice (*n* = 4–5 per time point). Splenic CD4+ and CD8+ T cells were isolated by magnetic cell sorting (MACS; negative selection) using mouse CD4+ and CD8a+ T cell isolation kits (#130-104-454 and #130-104-075, Miltenyi Biotec, Bergisch Gladbach, Germany) on autoMACS from Miltenyi Biotec following the manufacturer’s instructions. Ratios of CD4+ and CD8+ T cells were calculated from total splenic cell counts and normalized to the ratio of the CON group. The purity of isolated cells was verified by flow cytometry using anti-CD4 (Brilliant Violet 711, clone RM4-5, #100549) and anti-CD8a (Alexa Fluor 700, clone 53-6.7, #100729) antibodies (Biolegend, San Diego, CA, USA). The purity of CD4+ or CD8+ T cells was quantified as a percentage of single live lymphocytes. Mean purities were 82.4 ± 0.9% for CD4+ T cells and 74.1 ± 1.7% for CD8+ T cells.

### 2.4. Real-Time Quantitative Reverse Transcription PCR (qPCR)

Total RNA was extracted from isolated T cells with the RNeasy Mini Kit (#74104; Qiagen, Hilden, Germany) following the manufacturer’s protocol. RNA was transcribed into cDNA (#4368814, Applied Biosystems, Foster City, CA, USA). qPCR was performed using the Go Taq qPCR Master Mix (#A6002, Promega, Walldorf, Germany). Briefly, 5 µL of cDNA was mixed with 10 µL SYBR Green Pre-Mix and 5 µL target primer mix (1.4 µM, Eurofins Genomics GmBH, Ebersberg, Germany). The following amplification program was used on a Bio-Rad CFX96 cycler (Bio-Rad, Hercules, CA, USA): 5 min at 94 °C, 40 cycles of 15 s at 94 °C, 15 s at 60 °C, and 20 s at 72 °C, and final extension for 5 min at 72 °C. Primer sequences are provided in Appendix A. Relative expression ratios for each transcript were calculated based on individual primer efficiencies, as described in [44], using Eef1a1 as the reference gene, which shows a high abundance but no circadian variation in expression. The circadian average (i.e., the average over four time points) of each gene’s relative expression in the CON group was used for normalization. To account for time-of-day variations in gene expression, we calculated the ratio of the gene’s expression level at a specific time point in each treatment group to the average expression level of the same gene at the same time point in the control group.

### 2.5. Statistical Analysis

Circadian rhythmicity and amplitude analysis of time series data (cell counts and gene expression) was performed using the CircaSingle (cosinor) function of the CircaCompare package, version 1.0.0 [45] in the R environment (version 4.1), applying an alpha threshold of 0.05 (Appendix A). Differences in EAE score kinetics were analyzed by least-squares fitting sigmoidal dose response with variable slope curves and determining the time interval when 50% of the maximum EAE score (ET50) was reached for each single animal. Differences between EAE and CFA/PTx groups were estimated by unpaired two-tailored *t*-tests. Effects of treatment (CFA/PTx or EAE) against CON adjusted for time-of-day were assessed by one-sample *t*-tests against 1. Calculations were performed using GraphPad Prism 9 (GraphPad Software, La Jolla, CA, USA). *p*-values below 0.05 were considered significant. Error bars depict SEMs.

## 3. Results

### 3.1. Central Circadian Rhythms Are Largely Unaffected in EAE

To study how central (SCN) circadian clock-controlled functions are affected during EAE, we analyzed disease progression and locomotor activity rhythms in mice kept under rhythmic (LD) or non-rhythmic (DD) environmental conditions (Figure 1A). EAE animals developed first signs of paralysis around day 11 and reached a chronic (plateau) phase of the disease on days 16–18 after induction, while CFA/PTx-treated mice showed no disease symptoms (Figure 1B). 

Mice showed a marked spike in relative light phase activity (due to reduced activity levels in the first dark phase after the treatment) on the first day of immunization (experimental day 0) but reverted to a normal LD activity distribution on the next day (Figure 1C). A separate cohort of EAE mice was released into DD conditions starting on day 2 after immunization. EAE mice showed stable activity rhythms with free-running period lengths comparable to CON mice (EAE: 23.85 ± 0.06 h, CON: 23.81 ± 0.04 h; Figure 1D). Together, these data suggest that EAE disease progression has no significant effect on SCN-controlled locomotor activity rhythms.

### 3.2. Splenic T Cell Numbers Are Reduced in Response to CFA/PTx Treatment and during Chronic EAE 

To analyze potential effects of chronic inflammation on peripheral T cell rhythms, we sacrificed CFA/PTx, EAE, and CON animals on days 25–27 after treatment under DD conditions to avoid a potential impact of external time cues on gene expression, and we isolated splenic CD4^+^ and CD8^+^ T cells. Overall, abundances of both cell types showed little variation over the course of the day in all three experimental groups (Figure 2A,C). A significant circadian rhythm with a modest amplitude (ca. 15%) was detected specifically in CD4^+^ T cells of EAE mice (Figure 2A). When correcting for time-of-day effects, splenic CD4^+^ and CD8^+^ T cells were significantly lower in CFA/PTx and EAE compared to CON animals (Figure 2B,D). Interestingly, down-regulation was more pronounced in CFA/PTx than in EAE for both T cell populations (Figure 2B,D). 

### 3.3. Persistent Disruption of Circadian T Cell Clock Gene Rhythms in Response to CFA/PTx Treatment and EAE

Next, the expression profiles of clock genes in splenic CD4^+^ and CD8^+^ T cells obtained from mice in all three groups (EAE, CFA/PTx, and CON) were determined. We investigated the genes of the core molecular clock which have been shown to express high-amplitude rhythms in several tissues. We aimed to include clock genes from both the positive (*Bmal1*) and the negative (*Per2*) arm of the core TTFL, as well as the accessory loops (*Nr1d1* and *Dbp*). In CD4^+^ T cells of CON mice, all the tested clock gene mRNAs showed significant circadian rhythms, with the expected anti-phasic regulation between *Bmal1* and the E-box regulated *Dbp*, *Per1*, and *Nr1d1* (Figure 3A–D, Appendix A). In CFA/PTx mice, rhythms in two clock genes (*Bmal1* and *Dbp*) lost significance, but only *Dbp* showed a significantly decreased amplitude compared to CON (Figure 3A,C). *Per1* peak expression in CFA/PTx mice almost coincided with trough levels in CON mice, suggesting a phase inversion (Figure 3B,E). In EAE animals, rhythms in *Per1* and *Dbp* were lost, and *Dbp* showed a significantly lower amplitude (Figure 3B,C), while *Bmal1* expression was still rhythmic but almost anti-phasic compared to CON (Figure 3A,E). In contrast, *Nr1d1* mRNA rhythms were stable across all conditions, and phasing was largely preserved (Figure 3D,E). When correcting for time-of-day effects, expression levels of *Per1* and *Dbp* were overall decreased in the CFA/PTx group compared to both CON and EAE mice (Figure 3B,C). *Nr1d1* levels were similarly lowered in CFA/PTx and EAE mice (Figure 3D). *Bmal1* expression was significantly increased in CFA/PTx but not in EAE compared to CON mice (Figure 3A). 

In CD8^+^ cells of CON mice, *Per1* and *Dbp* showed significant circadian rhythms with comparable phase regulation, while *Bmal1* and *Nr1d1* profiles were not found to be significantly rhythmic (Figure 4A–D). In CFA/PTx mice, rhythms in *Dbp*, *Per1*, and *Nr1d1* were preserved, and *Dbp* showed a lower amplitude compared to CON. *Per1* and *Dbp* were phase-advanced by about 7.5 h compared to the rhythms of the CON group (Figure 4B,C,E). In EAE animals, none of the clock genes retained a rhythmic profile and, additionally, *Dbp* showed a significantly lower amplitude compared to CON. When correcting for time-of-day effects, *Per1* and *Dbp* expression were similarly reduced in CFA/PTx and EAE mice compared to the CON group. *Bmal1* mRNA levels, in turn, were consistently upregulated, which was more prominent in CFA/PTx mice (Figure 4A). *Nr1d1* expression exhibited a trend towards a decrease (*p* = 0.051) in CFA/PTx but not in EAE mice compared to the CON group (Figure 4D). 

In summary, we found that clock gene expression rhythms were consistently disrupted in splenic CD4^+^ T cells of CFA/PTx-treated and chronic-phase EAE mice. In CD8^+^, clock gene rhythms were preserved in CFA/PTx-injected mice but abolished in EAE animals. These results indicate a treatment- and cell-specific effect on the T cell clock function of chronic inflammation. 

### 3.4. Circadian Expression of Immune-Function-Related Genes in Peripheral T Cells

Clock gene machinery affects cellular functions in part through the rhythmic regulation of transcriptional targets [46]. As a potential indicator of how changes in T cell clock regulation in CFA/PTx and EAE mice may translate into functional alterations, we measured the mRNA level profiles of immune markers in CD4^+^ and CD8^+^ T cells. In splenic CD4^+^ T cells, we measured the expression of genes that encode transcription factors critical for T cell differentiation towards Th2 (*GATA-binding protein 3*, *Gata3*) and regulatory T (Treg) (*forkhead box P3*, *Foxp3*) as well as pro-inflammatory Th1 (*T-box 21*, *Tbx21*, encodes T-bet) and Th17 (*RAR related orphan receptor C*, *Rorc*, encodes RORγt) subsets, and two cytokines—*interleukin 10* (*Il10*) and *interferon gamma* (*Ifng*). *Gata3* and *Foxp3* showed rhythmic expression profiles in the CON group, with peak expression in the late rest/early active phase of the animals (46–49 h after “lights off”). These rhythms were disrupted in CFA/PTx and EAE mice with significant amplitude reductions in EAE mice (Figure 5A,B, left panels). When comparing expression levels adjusted for circadian time, *Gata3* expression was significantly upregulated in CFA/PTx but not in EAE mice (Figure 5A, right panel). *Foxp3* was downregulated in EAE compared to CON (Figure 5B, right panel). For both genes, expression levels were significantly lower in EAE compared to CFA/PTx mice (Figure 5A,B, right panels). The expression of *Il10* was not rhythmic in any of the conditions and downregulated in the CFA/PTx and EAE groups (Figure 5C). None of the analyzed pro-inflammatory transcripts showed rhythmic regulation in any of the cohorts (Figure 5D–F, left panels). The levels of *Tbx21* were higher in CFA/PTx compared to CON and EAE (Figure 5D, right panel). The *Ifng* time-of-day corrected levels were largely unaffected (Figure 5E, right panel), while *Rorc* transcription was reduced in CFA/PTx and EAE mice to a similar extent (Figure 5F, right panel). 

In CD8^+^ cells, we analyzed the transcript level profiles of *perforin 1* (*Prf1*) and *granzyme B* (*Gzmb*) as markers for the effector functions of cytotoxic T cells. We found a significant circadian oscillation of *Prf1* in CFA/PTx but not in EAE nor in CON conditions (Figure 6A, left panel). No rhythmic expression of *Gzmb* was observed in any condition (Figure 6B, left panel). For *Prf1*, time-of-day adjusted expression was higher in CFA/PTx and EAE relative to CON. The levels of *Prf1* were significantly higher in CFA/PTx compared to EAE mice (Figure 6A, right panel). *Gzmb* levels were significantly lower in CFA/PTx when adjusted for time-of-day than in CON (Figure 6B, right panel).

Taken together, *Gata3* and *Foxp3* expression levels showed a significant circadian variation in untreated mice. These rhythms were lost in CFA/PTx-treated and EAE-induced mice. Besides that, chronic inflammation mainly had time-of-day-independent effects on the expression of selected immune genes.

## 4. Discussion

Here, we report that CFA/PTx and CFA+MOG(35-55)/PTx (EAE) treatments disrupt circadian clocks in splenic CD4^+^ and CD8^+^ T cells. The expression of selected immune genes, in contrast, was mostly altered in a time-of-day-independent manner, suggesting T cell clock independent regulation. T cell circadian clock disruption is likely independent of central clock function, as evidenced by unaltered circadian rhythms of locomotor activity. 

We observed that circadian rhythms of locomotor activity were sustained in mice with chronic EAE under LD and DD conditions (Figure 1). Under rhythmic light conditions, mice were mostly active during the dark phase before immunization, while animals display a phase of lowered activity between ZT16 and ZT20 after immunization until the end of the experiment, which might indicate an overall weakening effect of the resulting inflammation. Importantly, after a transient immunization response on day 1, the distribution of light and dark phase activity remained similar between pre- and post-immunization (Figure 1), suggesting stable central circadian rhythmicity. Moreover, under DD conditions, the SCN-controlled activity period length in EAE mice was comparable to CON animals, indicating unperturbed SCN clock function. Hind paw inflammation induced by CFA injection reportedly also does not change circadian locomotor activity rhythms [47]. These results favor a scenario in which potential alterations of the central clock in chronic inflammation by CFA/PTx-treatment or EAE induction, if present at all [36,48], are subtle and insufficient to markedly alter circadian behavioral rhythms. However, changes in centrally regulated circadian output rhythms such as sleep architecture and heart rate over the course of the experiment may still be present [48,49]. 

Circadian rhythms in a number of immune cell subsets have been observed in blood, lymph nodes, and spleen, with distinct circadian patterns [13,16,19,50,51], while others found no circadian variation in numbers of specific T cell subpopulations in the lymph node [14]. These contradictory results indicate that these variations might be of low amplitude. In accordance with others [50], we also observed the highest abundance of CD4^+^ and CD8^+^ T cells in the spleen during the early active phase (49 h in DD; Figure 2). Proportions of splenic CD4^+^ and CD8^+^ T cells were lowest during the early inactive phase (37–43 h in DD). A circadian significant rhythm was only detected for the abundance of CD4^+^ T cells in EAE mice (Figure 2). In line with this, circadian rhythms in T cell counts within the spleen were found to be of low amplitude and high individual variance, which we also detected in our experiment [18,50]. Proportions of CD4^+^ and CD8^+^ T cells in spleen were overall reduced in CFA/PTx and EAE mice compared to CON. Increases in these T cell subsets are known to occur around the onset of EAE, but they revert to pre-immunization levels at the chronic stage [52]. However, the observed reduction to below pre-EAE levels was unexpected and requires further investigation. In a previous study, we have shown that the genetic deletion of intrinsic clocks affects the trafficking of peripheral T cells to lymph nodes and activation by antigen presenting cells (APCs) [19]. We here also observed a disruption of the circadian clock in CD4^+^ and CD8^+^ T cells, but it remains to be shown if this similarly affects migration in the spleen. 

Most immune cells have been shown to harbor intrinsic circadian clocks [16,18,22]. However, amplitudes of circadian gene oscillations are found to be small compared to solid tissues such as the liver [17], sometimes failing to reach significance with regard to rhythmic expression [16,17,18]. In our study, *Bmal1*, *Per1*, *Dbp*, and *Nr1d1* were rhythmically expressed in CON CD4^+^ T cells, while the significance of these rhythms was partially lost in chronic inflammatory state (Figure 3). Clock gene expression rhythms in CD4^+^ T cells were phase-shifted compared to published data from spleen [17], but they were similarly phased to those previously observed in lymph nodes [19]. In CD8^+^ T cells of CON mice, *Bmal1* and *Nr1d1* did not show significant rhythmicity, while *Per1* and *Dbp* oscillated in a circadian manner (Figure 4). Nobis and colleagues found that about 6% of protein-coding transcripts show circadian rhythmicity in CD8^+^ T cells isolated from lymph nodes, including *Nr1d1* and *Dbp*, whereas *Bmal1* and *Per1* did not show significant circadian rhythms [18]. However, amplitudes were usually low, and the researchers pointed out that the expression levels of several clock genes showed variance over the day but did not classify as rhythmic due to stringent cut-off criteria. The phasing of clock gene expression varies under LD and DD conditions [53], and this—together with differences in the T cell subsets studied—may explain some phase differences under control conditions between ours and some other studies [18,19]. Importantly, *Bmal1*, part of the activator arm of the circadian core TTFL, shows the expected anti-phasic relationship to *Per1* and *Dbp* (as members of the negative arm) in our study and others. The phasing of *Nr1d1* was similar in all groups in CD4^+^ T cells, while *Per1* in CFA/PTx-treated mice and *Bmal1* in EAE mice showed large phase differences to CON, hinting towards an overall intra-cellular desynchrony of the circadian clock. For some clock genes, we observed differences in expression levels between CFA/PTx and EAE mice. Importantly, the direction of change in expression was the same for both groups compared to CON. Clock gene expressions are impacted by complex, not well-studied mechanisms, including chromatin remodeling and post-transcriptional and as well as post-translational modifications [3,4,5], which should be addressed in future studies.

Our results add to the growing evidence that chronic inflammation disrupts circadian rhythms not only in various immune cells such as macrophages, neutrophils, and mononuclear cells but also in peripheral organs like liver, colon, or adipose tissue [36,37,38,39,40,41,42,43]. It will be important to investigate the physiological effects of such circadian disruption. Moreover, REV-ERBα has been described as an important integrator of circadian rhythms and energy metabolism [54], and the downregulation of *Nr1d1* expression may indicate altered fuel utilization in CD4+ T cells under chronic inflammatory conditions. 

Circadian clocks control the immune response in a time-of-day-dependent manner by regulating immune cell migration or cytokine production, and clock disruption induces or aggravates inflammation [30,55,56]. Here, we only found circadian rhythms in *Gata3*, highly expressed in Th2 cells, and *Foxp3*, a marker for Treg cells, in CON mice (Figure 5). *Gata3* and *Bmal1* expression peaked at comparable times, which is in line with the published data [57]. Overall, *Gata3* expression was increased in CFA/PTx, while *Foxp3* was downregulated in EAE mice. This may indicate that the proportion of corresponding splenic T cell subpopulations may differ between the treatment groups. Treg cells are reduced for at least 70 days after EAE induction. An effect mediated by PTx, however, was only investigated for up to 10 days [58]. Our data suggest that PTx may initiate this phenomenon, but it is negligible during chronic stage in EAE. *Gata3* overexpression inhibits the differentiation of naïve T cells into Th17, resulting in reduced *Rorc* expression [59]. CFA/PTx mice also exhibited increased *Gata3* and decreased *Rorc* expression. The deletion of the transcriptional repressor and clock component REV-ERBα inhibits Th17 development, increasing the pro-inflammatory state, including upregulated *Rorc* expression [60]. We observe decreased *Nr1d1* levels but no increase in *Rorc* expression in both treatment groups, which may indicate a lower abundance of Th17 cells or a secondary effect of overall decreased CD4^+^ T cell numbers in those groups compared to CON mice. *Il10* expression was time-of-day-independently lowered during chronic inflammation, as previously reported [52]. *Tbx21* expression was upregulated in CFA/PTx but not EAE mice. However, this did not result in increased *Ifng* levels, which is in contrast to Sonobe et al.’s findings [52]. Of note, we observed higher expression of *Ifng* in CFA/PTx as well as in EAE mice at certain times of day. Thus, it cannot be ruled out that discrepancies arise due to sampling times, which was not indicated in their study [52]. In CD8^+^ T cells, two of the main effector genes, *Gzmb* and *Prf1*, showed a trend for rhythmicity (*p* = 0.088 and 0.087, respectively) in CON. This is consistent with a previous report in CD8^+^ T cells extracted from lymph nodes [18]. Interestingly, *Prf1* expression became rhythmic in CFA/PTx mice. This effect, however, is unlikely to be a direct effect of the circadian clock since the phase relationship of clock gene rhythms was disrupted in CFA/PTx compared to CON mice. Overall, *Prf1* levels were higher in both treatment groups, whereas *Gzmb* expression was decreased in CFA/PTx only. Future studies may elucidate to which extent these effects are mediated by disruptions of systemic circadian signals such as glucocorticoids [61,62]. 

This study has several limitations. For one, we only investigated gene expression in splenic CD4^+^ and CD8^+^ T cells. Further experiments are needed to evaluate if T cells in other lymphoid tissues are similarly affected. Further, we cannot exclude that part of the observed alterations in clock gene expression might be mediated by changes in food and water intake or social behavior rhythms in treated mice [63]. However, reductions in food and water intake are mainly seen during the acute phase, but they normalize—together with body weight—during the chronic phase of EAE [64]. Moreover, several studies have shown that activity and food intake rhythms strongly correlate [65,66,67], and our IR activity data suggest stable circadian rhythms in locomotor activity in treated mice. 

Another limitation of our study is based on the fact that the potency of PTx preparations may change between lots, and growing evidence stresses the importance of adjusting the amount of injected PTx to achieve comparable disease outcomes [68]. Different PTx lots were used for the EAE and CFA/PTx cohorts. We tried to adjust for different potencies in our study but cannot exclude that effect size differences between the treatment groups may arise from uncompensated differences in potency. Nonetheless, our data in terms of T cell abundance and gene transcription hint towards a long-lasting combinatorial effect of CFA depots and PTx treatment [69]—at least in spleen. 

Finally, in our experimental setup, we cannot discriminate whether effects on circadian gene expression are directly mediated by inflammatory markers, secondary to other systemic effects, or changes in the microenvironment, which may be elucidated in further studies. We also acknowledge that the low time resolution of circadian profiles prevents more detailed analysis of circadian rhythm characteristics.

## 5. Conclusions

In conclusion, we show that chronic inflammation, induced by CFA/PTx injection or during late-stage EAE, disrupts molecular clock rhythms in splenic CD4^+^ and CD8^+^ T cells in a cell-specific manner. This clock disruption occurs largely independently of central clock function and is not mimicked by the changes observed in immune gene expression. These findings support growing evidence on the disruptive capacity of chronic inflammatory states on peripheral circadian rhythms. Further studies are needed to evaluate to which extent disrupted T cell clock gene rhythms are manifested in other models of chronic inflammation and if these effects apply similarly to other cells and functions of adaptive immunity. Further studies are needed to elucidate the physiological effect of clock disruption, e.g., changes in T cell infiltration into the nervous system.

## Figures and Tables

**Figure 1 cells-13-00151-f001:**
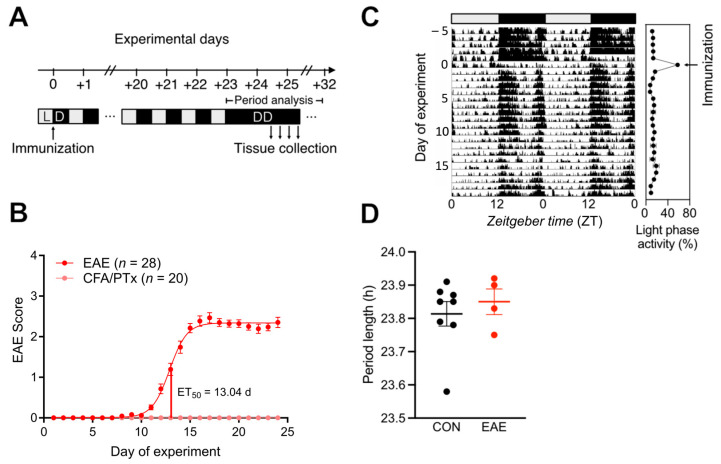
EAE expression has no lasting effect on the rhythmicity of locomotor activity. (**A**) Experimental setup (LD—12 h light:12 h dark cycle; DD—constant darkness). (**B**) EAE symptom dynamics in EAE (red, *n* = 28) and CFA/PTx only-treated (pink, CFA/PTx, *n* = 20) mice in LD. (**C**) Representative actogram of group-housed mice using passive infrared detection in LD (left panel) and average light phase activity (relative to total activity before immunization; right panel) during the experiment. (**D**) Free-running activity period length of CON (black) and EAE mice upon release into DD (data points indicate number of cages). Error bars depict SEM.

**Figure 2 cells-13-00151-f002:**
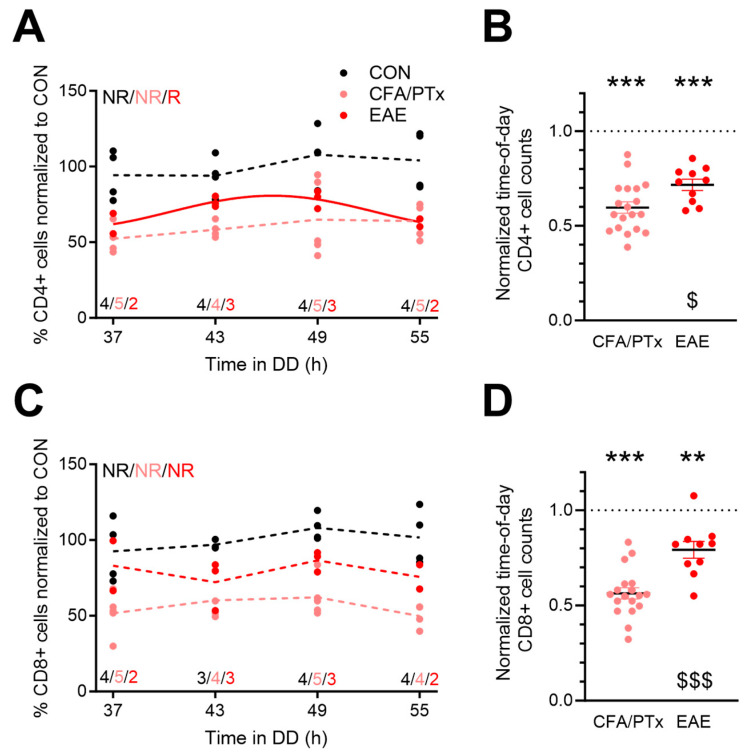
Splenic T cell numbers are decreased after CFA/PTx treatment and in EAE with little influence on circadian time. (**A**,**C**) CFA/PTx and EAE circadian profiles of splenic (**A**) CD4^+^ and (**C**) CD8^+^ T cells relative to CON (per time point: *n* = 3–4 (CON, black), *n* = 4–5 (CFA/PTx, pink), *n* = 2–3 (EAE, red)). (**B**,**D**) Time-of-day adjusted changes in (**B**) CD4^+^ and (**D**) CD8^+^ T cells in CFA/PTx (CD4^+^: *n* = 19, CD8^+^: *n* = 18) and EAE (*n* = 10). Mice were sacrificed at 6 h intervals on the second day of DD. Letters in (**A**) and (**C**) indicate results from cosinor analysis (R—rhythmic (*p* < 0.05; see Appendix A), NR—non-rhythmic). Sine waves were plotted for significantly rhythmic profiles indicated by solid lines; hashed lines indicate non-rhythmic profiles. **: *p* < 0.01 ***: *p* < 0.001; one sample *t*-test against 1 (CON group). $: *p* < 0.05; $$$: *p* < 0.001; unpaired two-tailed *t*-test (CFA/PTx vs. EAE). Error bars depict SEM.

**Figure 3 cells-13-00151-f003:**
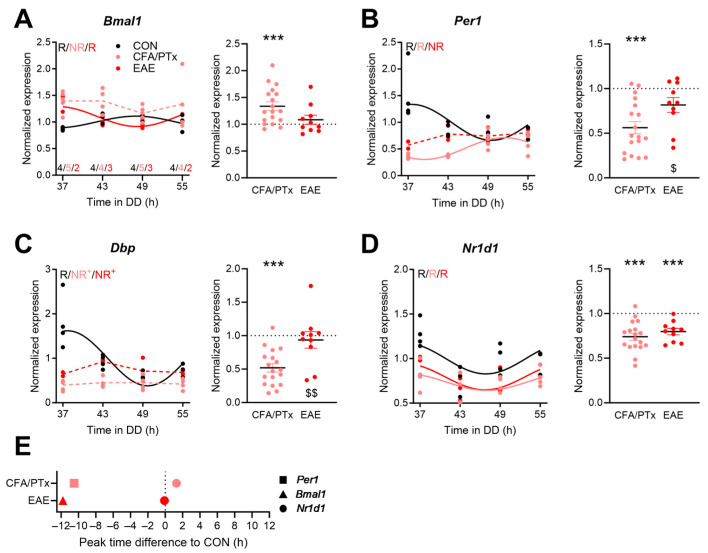
Clock gene expression rhythms are disrupted in splenic CD4^+^ T cells after CFA/PTx treatment and in EAE. (**A**–**D**) Circadian profiles (left panels) and time-of-day adjusted clock gene mRNA levels (right panels) in splenic CD4^+^ T cells in CFA/PTx, and EAE mice relative to the CON group. Mice were sacrificed at 6 h intervals on the second day of DD (left panels, per time point: *n* = 3–4 (CON, black), *n* = 4–5 (CFA/PTx, pink), *n* = 2–3 (EAE, red) as indicated in (**A**); right panels: *n* = 18 (CFA/PTx), *n*= 10 (EAE)). (**E**) Shifts in expression peak phase of rhythmic clock genes in EAE and CFA/PTx mice relative to CON. Letters in (**A**–**D**) indicate results from cosinor analysis (R—rhythmic, NR—non-rhythmic). Sine waves were plotted for rhythmic profiles indicated by solid lines; hashed lines indicate non-rhythmic profiles. + indicates changes in amplitude compared to CON, evidenced by cosinor analysis. ***: *p* < 0.001; one sample *t*-test against 1. $: *p* < 0.05; $$: *p* < 0.01; unpaired two-tailed *t*-test. Error bars depict SEM.

**Figure 4 cells-13-00151-f004:**
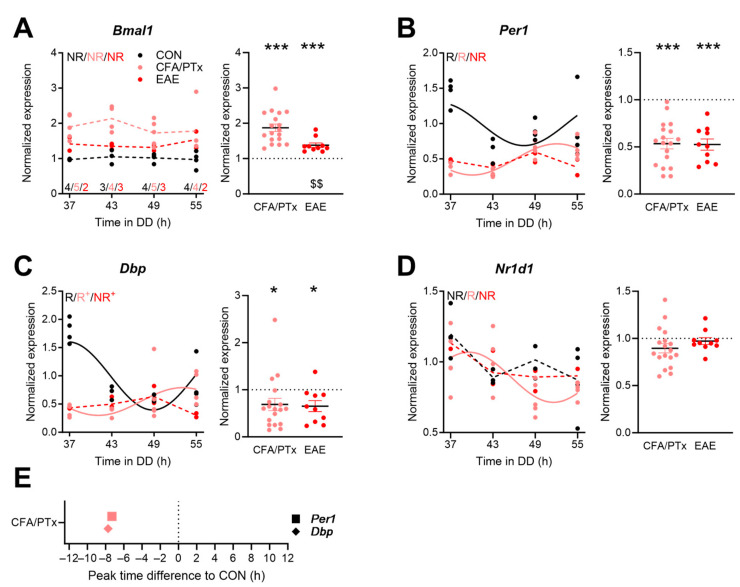
Clock gene expression rhythms are disrupted in splenic CD8^+^ T cells after CFA/PTx treatment and in EAE. (**A**–**D**) Circadian profiles (left panels) and time-of-day adjusted changes in clock gene mRNA levels (right panels) in splenic CD8^+^ T cells in CFA/PTx, and EAE mice relative to the CON group. Mice were sacrificed at 6 h intervals on the second day of DD (left panels, per time point: *n* = 3–4 (CON, black), *n* = 4–5 (CFA/PTx, pink), *n* = 2–3 (EAE, red), as indicated in (**A**); right panels: *n* = 18 (CFA/PTx), *n*= 10 (EAE)). (**E**) Shifts in expression peak phases of rhythmic clock genes in CFA/PTx mice relative to CON. Letters in (**A**–**D**) indicate results from cosinor analysis (R—rhythmic, NR—non-rhythmic). Sine waves were plotted for rhythmic profiles indicated by solid lines; hashed lines indicate non-rhythmic profiles. + indicates changes in amplitude compared to CON, evidenced by cosinor analysis. *: *p* < 0.05, ***: *p* < 0.001; one sample *t*-test against 1. $$: *p* < 0.01; unpaired two-tailed *t*-test. Error bars depict SEM.

**Figure 5 cells-13-00151-f005:**
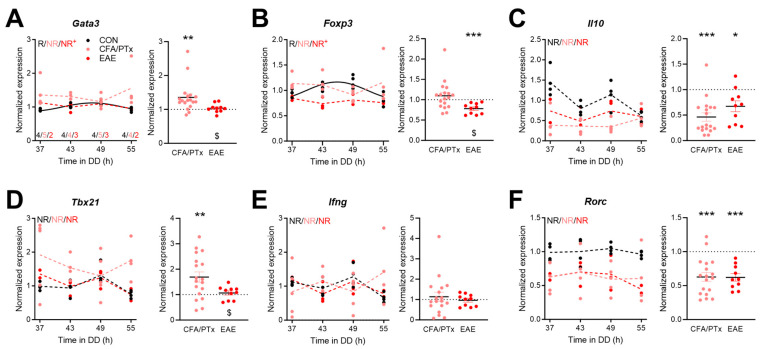
Anti-inflammatory immune cell marker gene expression rhythms are disrupted and *Rorc* expression level is decreased in splenic CD4^+^ T cells after CFA/PTx treatment and in EAE. (**A**–**F**) Circadian profiles (left panels) and time-of-day adjusted changes (right panels) of mRNA levels of (**A**–**C**) anti-inflammatory and (**E**,**F**) pro-inflammatory genes in splenic CD4^+^ T cells in CFA/PTx and EAE mice relative to the CON group. Mice were sacrificed at 6 h intervals on the second day of DD (left panels, per time point: *n* = 3–4 (CON, black), *n* = 4–5 (CFA/PTx, pink), *n* = 2–3 (EAE, red), as indicated in (**A**); right panels: *n* = 18 (CFA/PTx), *n*= 10 (EAE)). Letters in (**A**–**F**) indicate results from cosinor analysis (R—rhythmic, NR—non-rhythmic). Sine waves were plotted for significantly rhythmic profiles indicated by solid lines; hashed lines for non-rhythmic profiles. + indicates changes in amplitude compared to CON, evidenced by cosinor analysis. *: *p* < 0.05, **: *p* < 0.01, ***: *p* < 0.001; one sample *t*-test against 1. $: *p* < 0.05; unpaired two-tailed *t*-test. Error bars depict SEM.

**Figure 6 cells-13-00151-f006:**
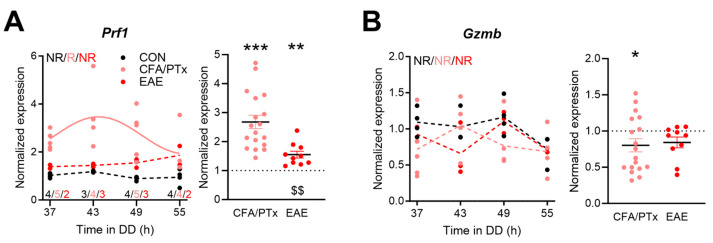
*Prf1* and *Gzmb* mRNA expressions are regulated in opposite directions in splenic CD8^+^ cells after CFA/PTx and EAE treatment. (**A**,**B**) Circadian profiles (left panels) and time-of-day adjusted changes in mRNA levels (right panels) of (**A**) *Prf1* and (**B**) *Gzmb* in splenic CD8^+^ T cells in CFA/PTx, and EAE mice relative to the CON group. Mice were sacrificed at 6 h intervals on the second day of DD (left panels, per time point: *n* = 3–4 (CON, black), *n* = 4–5 (CFA/PTx, pink), *n* = 2–3 (EAE, red), as indicated in (**A**); right panels: *n* = 18 (CFA/PTx), *n*= 10 (EAE)). Letters in (**A**,**B**) indicate results from cosinor analysis (R—rhythmic, NR—non-rhythmic). Sine waves were plotted for significantly rhythmic profiles indicated by solid lines; hashed lines indicate non-rhythmic profiles. *: *p* < 0.05, **: *p* < 0.01, ***: *p* < 0.001; one sample *t*-test against 1. $$: *p* < 0.01; unpaired two-tailed *t*-test. Error bars depict SEM.

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
