# Peer review of "Chronic Inflammation Disrupts Circadian Rhythms in Splenic CD4+ and CD8+ T Cells in Mice"

_cells, 2024, doi:10.3390/cells13020151_

Round 1

Reviewer 1 Report

Comments and Suggestions for Authors

The manuscript entitled “Chronic inflammation disrupts circadian rhythms in splenic CD4+ and CD8+ T cells in mice” by Hirose et al. is an excellent evaluation of the impact of chronic inflammation, instigated from two separate sources, and its impact on splenic CD4 and CD8 T cells. This work fills a gap in the current understanding of how chronic inflammation impacts the adaptive immune response whereas other aspects, such as the circadian rhythm in the innate response, has been well investigated. The group uses an excellent and well-validated model to address the impact. The group had interestingly found that the impact of this inflammation varied by cell type, and this variation was further identified through different sources of inflammation. Another very interesting finding is that the transcription factors responsible for T cell differentiation were also impacted by this inflammation. Finally, the finding that the effector functions of these were, by and large, not impacted by this. It is a well designed and well executed study. Most of the limitations were addressed in the discussion, with the only exception being a slightly lower purity of T cell isolation that would be expected. However, as significant changes were still observed, and differences between the groups was clearly obvious, increased purity may not be necessary. Great job!

Author Response

We thank the reviewer for the positive review. We agree that purity of individual T cell populations was slightly lower than expected but differences in expression levels were explicit. Thus, we are confident that this does not significantly affect our conclusions.

Reviewer 2 Report

Comments and Suggestions for Authors

In this study, the authors check if the chronic Inflammation disrupts circadian rhythms. This is very important and can influence the immune functions and disease outcome. To study the impact of chronic inflammation on the internal circadian clocks using experimental autoimmune encephalomyelitis (EAE) for multiple sclerosis. The manuscript is will organized and written, however there are few modifications are needed

.

Major comments

1-     In the introduction, the author should mention why the late-stage EAE don’t influence the circadian rhythms?

2-     In the introduction, the authors need to give more information about the genes which influencing circadian clocks.

3-     One which basis you choose the genes Bmal1, Dbp, Eef1a…etc.? This should be mentioned in the introduction or in the result part.

4-     Could you explain the difference in the gene expression (for Per1 and Dbp, Nr1d1 and Bmal1) between the EAE group and CFA/PTx group? Why the EAE group which has a higher EAE score showed less gene expression difference compared to CFA/PTx group? CD4+ and CD8+ T cells were also significantly lower in CFA/PTx, could you explain why?

Minor comments

1-     In the introduction, describe the abbreviation at the first appearance (ARNT, BMAL1,….). When the abbreviation is mentioned, there is no need to mention the full name again (line 49 it is enough to write EAE)

2-     Line 36-43 what you write needs support with references

3-     In the introduction, no need to cite the results, delete the line 70-75 (We found…. clock-independent regulation)

4-     In material and methods, for all reagents, instrument and software, you need to mention the company, city and county.

5-     In material and methods, mention the difference between all group and the treatment used. What are the characteristics for late-stage EAE, and CFA/PTx-treated groups

6-     In the results, for the diagram, the legend must be mentioned and the number of replicates. Each group must have same legend in all diagram. Example, in Figure 2 A and C, the legend for EAE group are different, one time continuous (A) the other dashed line (C). For figure 3 also different legend (figure 3 A and B) for the same group.

Comments on the Quality of English Language

Minor editing of English language required

Author Response

We would like to thank this reviewer for the thorough evaluation of and valuable feedback on our manuscript. We addressed the comments point-by-point:

Major comments

  • In the introduction, the author should mention why the late-stage EAE don’t influence the circadian rhythms?

Answer: We did not include a detailed description of the impact of EAE on circadian rhythms in the introduction since the focus of the study was on transcriptional changes in peripheral T cells. However, we highlight that coherent circadian rhythms in behavior and physiology are driven by the SCN (lines 53-55). Buenafe ([36], doi:10.1016/j.jneuroim.2011.12.002) pointed out that chronic inflammation may directly or indirectly impact the SCN which, in turn, may cause disruption of circadian rhythms. The researcher only indirectly studied this by evaluation of hormonal rhythms, which are, inter alia, regulated by the SCN. Of note, samples were collected 2-8 days post disease onset and results may not completely reflect the situation in chronic EAE. In MS patients, reports on hormonal rhythms, such as melatonin, and sleep disorders (which may be caused by disruptions in the SCN) are contradictory (Tonetti er al., doi: 10.3390/jcm8122216; Taphoorn et al., doi: 10.1007/BF00867360 ; Kern et al., doi: 10.1007/s00109-019-01821-w). In our study, we investigated activity rhythms under different light conditions and found no changes between the treatment groups (EAE and CFA/PTx) and naïve mice which is in line with Pitzer et al. ([47], doi:10.1097/PR9.0000000000000564) and suggests that the SCN clock is, if at all, only subtly affected. For further details, we would like to refer to the discussion part (lines 338-353). A comprehensive analysis of alterations in circadian rhythms and in-depth study of possible disruptions in the SCN is missing to date and, moreover, may require large cohort sizes due to heterogeneous disease outcome.

2-     In the introduction, the authors need to give more information about the genes which influencing circadian clocks.

Answer: We are aware of the complex mechanisms of the circadian clock system. In the revised manuscript, we elaborate on the molecular clock feedback loops. Furthermore, transcription and clock protein abundance is further regulated by chromatin remodeling, post-transcriptional and post-translational modifications including phosphorylations. We added this information in the introduction. (lines 35-50). In brief, the circadian clock is comprised of the positive arm consisting of BMAL1 and CLOCK forming heterodimers inducing Per1-3 and Cry1/2 expression as well as many other clock-controlled genes, e.g., Rev-Erbs, Rors, and Dbp. In turn, REV-ERBs and RORs inhibit and activate the expression of Bmal1, repectively. DBP binds to D-box promotor elements activating the expression of Pers, Crys, Rev-Erbs, and Rors. For a detailed description we would like to refer to the comprehensive reviews by Partch et al. ([2], doi:10.1016/j.tcb.2013.07.002), Takahashi ([1], doi:10.1111/dom.12514), Grimaldi et al. ([3], doi:10.1101/sqb.2007.72.049), Gallego ([4], doi:10.1038/nrm2106.), and Kojima et al., ([5], doi:10.1242/jcs.065771)).

3-     One which basis you choose the genes Bmal1, Dbp, Eef1a…etc.? This should be mentioned in the introduction or in the result part.

Answer: Thank you for this question. The chosen genes, Bmal1, Per2, Dbp and Nr1d1, are part of the core molecular clock, and they have been shown to express high-amplitude rhythms in several tissues. We also aimed at including genes from both the positive (Bmal1) and the negative arm of the core TTFL (Per2) as well as the accessory loops (Nr1d1 and Dbp) (lines 217-221). In this way, investigation of the expression level of those genes enables us to estimate the effect of chronic inflammation on the circadian clock machinery within CD4+ and CD8+ T cells. Of note, as pointed out in the discussion, NR1D1 is an important integrator not only in the circadian clock but also energy metabolism (lines 409-412). Eef1a is a commonly used reference gene in the circadian field which shows a high abundance and, importantly, does not exhibit a circadian variation in its expression. We added this information in the qPCR subsection (line 150): “Relative expression ratios for each transcript were calculated based on individual primer efficiencies as described [44] using Eef1a1 as reference gene, which shows a high abundance but no circadian variation in expression.”

4-     Could you explain the difference in the gene expression (for Per1 and Dbp, Nr1d1 and Bmal1) between the EAE group and CFA/PTx group? Why the EAE group which has a higher EAE score showed less gene expression difference compared to CFA/PTx group? CD4+ and CD8+ T cells were also significantly lower in CFA/PTx, could you explain why?

      Answer: We thank the reviewer for this comment. There is no simple answer to this, but it seems clear from our data that EAE score itself does not correlate with clock gene expression. Likewise, T cell numbers appear not to correlate with clock gene expression. We observe lower Nr1d1 time-of-day adjusted level in CFA/PTx and EAE animals compared to CON mice in CD4+ T cells. However, we did not detect any statistically significant difference between CFA/PTx and EAE. As discussed, there is evidence that Th17 cells show higher Nr1d1 expression compared to other CD4+ T cell subpopulations, and thus lower Nr1d1 levels may indicate decreased numbers of Th17 cells. However, a secondary effect caused by overall reduced CD4+ T cell numbers compared to CON group cannot be excluded (lines 427-432, [52],doi:10.1016/j.celrep.2018.11.101). It is not yet studied whether other clock genes exhibit differential expression patterns in other T cell subpopulations. As pointed out earlier, chromatin remodelling, post-transcriptional and post-translational (e.g., phosphorylation) modifications impact gene expression levels through complex mechanisms. These would be interesting to study in follow-up experiments. We added these points to the discussion (lines 399-404).

Minor comments

  • In the introduction, describe the abbreviation at the first appearance (ARNT, BMAL1,….). When the abbreviation is mentioned, there is no need to mention the full name again (line 49 it is enough to write EAE)

Answer: We changed the text as suggested.

  • Line 36-43 what you write needs support with references

Answer: We elaborated on molecular clock network as decribed above. We would like to point out that we cite comprehensive reviews on the molecular architecture of the circadian clock by Partch et al. ([2], doi:10.1016/j.tcb.2013.07.002) and Takahashi ([1], doi:10.1111/dom.12514). We highlighted these in the corresponding section. We furthermore briefly describe that the clock network is also regulated by chromatin remodelling, post-transcriptional and post-translational modifications for which we included references as well (Grimaldi et al. ([3], doi:10.1101/sqb.2007.72.049), Gallego ([4], doi:10.1038/nrm2106.), and Kojima et al., ([5], doi:10.1242/jcs.065771)).

  • In the introduction, no need to cite the results, delete the line 70-75 (We found…. clock-independent regulation)

Answer: We shortened this section. It now reads: “We found that not only EAE immunization, but also CFA/PTx treatment disrupted T cell circadian clocks while immune cell expression was mostly impacted in a non-circadian manner.”

  • In material and methods, for all reagents, instrument and software, you need to mention the company, city and county.

Answer: We added the missing information in materials and methods sections.

  • In material and methods, mention the difference between all group and the treatment used. What are the characteristics for late-stage EAE, and CFA/PTx-treated groups

Answer: We thank the reviewer for his comment. We included more details for each treatment group. In short, all animals were 9-13 weeks of age at the start of the treatment. We used age-matched CON animals which did not receive any injections. EAE animals reach a plateau in disease scores around day 18. From this day onwards only minor changes in diseases symptoms can be observed. This stage is known as chronic or plateau phase of EAE. We added further information of the EAE characteristic in the tissue collection and preparation subsection. CFA/PTx and CON animals do not develop any disease symptoms which can be validated by EAE score system. To have comparable time points to the EAE group we sacrificed mice of these groups at comparable days as EAE animals as pointed out in the subsection tissue collection and preparation. 

  • In the results, for the diagram, the legend must be mentioned and the number of replicates. Each group must have same legend in all diagram. Example, in Figure 2 A and C, the legend for EAE group are different, one time continuous (A) the other dashed line (C). For figure 3 also different legend (figure 3 A and B) for the same group.

Answer: We included the number of replicates in each figure legend. The line pattern is based on the cosinor analysis p-value results. Significantly rhythmic profiles are indicated by solid lines and non-rhythmic profiles are indicated by hashed lines. We added this information in the legends of Fig. 2-6.

Reviewer 3 Report

Comments and Suggestions for Authors

Hirose and colleagues describe how a chronic inflammation disrupts the circadian rhythm in murine splenic T cells. They report that during chronic EAE phase or after CFA/PTx treatment the expression level or the circadian oscillation of several clock genes are disrupted in CD4+ and CD8+ T cells isolated from spleen. However, this is not associated with rhythmic alterations in the expression level of key T cell markers, including Tbet, Ifng, Il10, Rorc, Prf, Grzb for which only total levels are altered compared to control mice. On the contrary, the authors observed that circadian oscillations of Foxp3 and Gata3 mRNAs are disrupted in EAE and CFA/PTx mice (and Foxp3 levels are also globally downregulated in EAE mice).

Overall, this study provides a first hint into the deregulation of clock genes in splenic T cells during chronic inflammation (especially interesting for the EAE model). I would have appreciated a more detailed characterization of the potential effects of the disruption of daily oscillations in circadian genes in T cells, especially for the EAE model. For example, in addition to the mRNA analyses, I am wondering if the authors could add an analysis of the infiltration of T cells into the nervous system to see if circadian variations in the number of infiltrating T cells are observed and if this correlates with mRNA data (especially for Tregs, given the loss of Foxp3 rhythm).

Could the authors repeat these analyses (maybe only on the EAE model and only for genes significantly altered in spleen) on T cells isolated from lymph nodes to see if the effect is limited to spleen or not?

Author Response

We would like to thank the reviewer for the positive assessment of our manuscript and for the valuable suggestions for future functional experiments on T cells.

Reply: We thank the reviewer for his comment. We agree that future studies are needed to elucidate the physiological impact of disrupted circadian rhythms in T cells such as alterations in T cell infiltration into the nervous system. We added a note on this direction in the end of the discussion. In our study, we focussed on whether chronic inflammation has disruptive effects on circadian clocks and rhythms in cells of adaptive immune system and specifically in splenic CD4+ and CD8+ T cells. Growing evidence on the clock-disruptive effect of inflammation on various cell types suggest that the effects we observe on splenic T cells are not limited to this organ but may also occur in other tissues such as the lymph nodes. We feel that conducting such extensive experiments in different T cell populations falls beyond the scope of the current study.  

Round 2

Reviewer 2 Report

Comments and Suggestions for Authors

The author needs to mention the city and country for „Hooke Laboratories”

The author adds the number of replicates in the figure legends, however, it is better to add it in the diagram.

Author Response

Reviewer 2 highlighted the missing country for Hooke Laboratories and suggested to add the number of replicates in the graphs. We followed these suggestions. With regard to the number of replicates, we added these in graphs where we show the data as means only, i.e. EAE score graph, or aligned dot plots, i.e. circadian profiles. Since replicate numbers for figures 3-6 are constant across panels, we only list them in the panel (A) of each figure.

Reviewer 3 Report

Comments and Suggestions for Authors

The authors did not perform any of the requested experiments.

Author Response

Reviewer 3 asked for additional experiments to evaluate the inflammatory effects in T cells isolated from lymph nodes. We agree with the reviewer, that it will be very interesting to investigate if the alterations observed in the spleen also translate to other lymphoid organs. However, we did not collect these for logistic reasons during past experiments. Further, these data would not interfere with conclusions drawn from our data so far. Thus, we consider these as potential future experiments. Our main aim for this study was to focus on the spleen and splenic T cells. We are aware, that our conclusions cannot be generalized to other lymphoid organs and, therefore, carefully avoided generalizations with this regard. We also revised the manuscript adding this point as a limitation of the study in the discussion part (lines 436-438, 452).